# Modeling of Magnetorheological Elastomers Using the Elastic–Plastic Model with Kinematic Hardening

**DOI:** 10.3390/ma12060892

**Published:** 2019-03-18

**Authors:** Daniel Lewandowski

**Affiliations:** Department of Mechanics, Material Science and Engineering, Wroclaw University of Science and Technology, 50-372 Wroclaw, Poland; daniel.lewandowski@pwr.edu.pl; Tel.: +48-602-75-13-45

**Keywords:** smart magnetic materials, magnetorheological composites, magnetorheological effect, mathematical models

## Abstract

The paper describes the modeling process of the magnetomechanical properties of magnetorheological elastomers. Unlike the primary sources in which a viscoelastic model is commonly used, in this work, the elastic–plastic model with linear hardening was adopted. Its parameters were determined from data obtained in an experiment. Measurement data encompassed a range of various strain rates and amplitudes, as well as a range of magnetic field values. Model parameters as a function of a magnetic field were obtained as a result of identification. The correctness of data correlation was shown by comparing hysteresis loops τ(γ). Satisfactory consistency between experimental and model research was achieved on the assumption that the model was applied only to higher strain rates, above the boundary value ${\dot{\gamma }}_{A}=0.025\;\left(1/s\right)$.

## 1. Introduction

The subject matter of this work is the analysis of the properties of a selected magnetorheological elastomer using a mathematical model. The material chosen for the study belongs to a large group of materials in the literature referred to as Smart Materials. Their essential characteristic is the ability to change their properties and parameters resulting from the effect of an external magnetic field [1,2]. In the case of magnetorheological composites and elastomers belonging to the same group of materials, mainly the change of their mechanical properties occurs. Due to their intended use, e.g., in mechanical constructions, mainly stiffness changes [3] or an ability to dissipate energy and damp vibration [4,5] arouse interest. It is the dissipation energy mechanisms of these materials, controlled and changed by a magnetic field, that allow using them in this way. They can also be used as actuators, which requires taking into account their magnetostrictive properties [6]. Stimulation with a magnetic field allows for a continuous change of material parameters. This characteristic means that it is possible to use magnetorheological elastomers in adaptive machines and equipment which can conform to changing operating conditions. However, this way of using materials requires the knowledge of their characteristics, i.e. a model. The description of the constitutive equation, the dependence of stress and strain as a function of a magnetic field, allows correctly constructing devices with these materials. The model is of crucial significance here also from the perspective of control and changes in a given construction. An example to illustrate this is vehicle suspensions that require conformity with current driving conditions, driving style and loads. The implementations of a magnetorheological fluid in semi-active control are presented in paper [7]. An example of using the Bouc–Wen model to describe the operation of dampers with a magnetorheological fluid in a car suspension can be found in [8].

The analysis of literature data on manufacturing, testing and modeling the properties of magnetorheological elastomers shows that a model that is frequently used in their description is the linear, viscoelastic Kelvin–Voight model [9,10]. Its popularity may be explained by the fact that it is commonly applied in the rheology of plastics where the warp of a magnetorheological elastomer is made of a polymer [11]. Unfortunately, this model is substantially simplified and does not allow correctly projecting some types of behavior, e.g., the influence of strain rate. In some scientific publications presenting research results related to these materials, their authors avoid showing stress and strain as a function of time. They offer only the so-called averaged analysis with a storage and loss modulus [12,13,14]. Unfortunately, these parameters are varied depending on frequency excitation, which is a mistake.

The proper description of magnetorheological elastomers with the use of the Kelvin–Voight model is related to their internal structure. Their crucial component is a ferromagnetic filling in the form of a magnetically soft powder, e.g., carbonyl iron [2] ASC 300 [15,16]. Mutual interaction between metallic particles with sizes of a few micrometers [17] is of crucial influence on its global properties, called the magnetorheological effect. It is generally defined as the dependence of stiffness and energy damping as a function of a magnetic field. Using the viscous element occurring in the Kelvin–Voight model as the factor responsible for an increase in stress linearly dependent on deformation speed is not sufficient. A correct description also requires the use of other compounds describing energy dissipation inside a material. A possible solution is the use of ideal-plastic elements [18,19].

This paper is divided into sections. In the initial Section 2, the MRE material and testing samples are discussed. Next, the description of the measurement stand and the test procedure are presented in Section 3. The most important part of the article is the model and process for the identification of its parameters, taking into account the influence of the magnetic field—Section 5 and Section 6. The work was completed by comparing the hysteresis loops obtained from the model and experimental data.

## 2. Magnetoreheological Elastomer and Specimens

The type of magnetorheological elastomer used and analyzed in this work was developed a few years ago. It consists of two main components: polymer matrix made of thermoplastic elastomer SEBS and a ferromagnetic filling—ASC300 iron type. Its detailed manufacturing procedure was described in a previous work by the author [20], where the experimental research of some properties of this type of materials was conducted. For the analysis of the model investigated in this work, a separate group of cylindrical shape samples with a height of 25 mm and a diameter of 25 mm was made. In the manufacturing process, the material was not polarized with an external magnetic field, and its internal structure can be considered anisotropic.

## 3. Test Stand and Magnetic Field Stimulation

The data used to identify model parameters were recorded during an experiment specially prepared for this purpose. A sample of a magnetorheological elastomer was cyclically loaded with simultaneous magnetic field simulation. Loading was conducted using a hydraulic device with a feedback system, which allowed obtaining a set of constant strain rate values. The magnetic field was generated using the so-called Halbach matrix constructed using a system of permanent magnets [21]. The matrix scheme and the method used to place the tested sample inside it are presented in Figure 1, showing: (1) the internal magnetic ring (fixed); (2) the external magnetic ring (rotating); (3) the cylindrical material sample; (4) the movable top plate attached to the sample (to transfer external force); and (5) the base. Red and blue vectors indicate action directions inside particular magnets, while the yellow ones represent a resultant magnetic field inside the matrix. The construction of the two-ring array allowed obtaining a directed magnetic field whose intensity could be adjusted by hanging the rotation angle. The measurement of magnetic field strength was conducted inside the matrix. The obtained characteristic is presented in Figure 2b. A change of the rotation angle between the rings causes the strengthening or weakening of the magnetic field inside it. This is due to the total sum of the magnetic field of all individual magnets that make up the matrix. This way of loading the sample and applying the magnetic field was described in detail in one of the previous works of the author [22].

## 4. Experimental Data

The samples made of a magnetorheological elastomer were deformed in a process similar to pure shearing. As shown in Figure 1, the top surface was moved parallel to the basis, and the investigated material was attached between them. The specimens were glued to the plates with a cyanoacrylate glue dedicated to rubber bonding—Loctite 480. The displacement measurement allowed calculating shear transformation in the form of angle γ. Stress values were calculated from the measurements of a force necessary to move the top surface and the sample cross-section. The character of deformation was a linear function cyclically changing at the constant speed. A sample waveform is presented in Figure 3. Hysteresis loops in the τ(γ) coordination system resemble an ellipsis; however, they have a sharp end and a straight line segment. Figure 4a shows hysteresis loops obtained for various frequency values (hence, also strain rates $\dot{\gamma }$) at a constant deformation amplitude and Figure 4b presents different deformation amplitude values at a constant strain rate.

## 5. Selection of Material Model

The model was selected based on the earlier experiments of the author and literature studies. The commonly used linear viscoelastic model, also called the Kelvin–Voigt model, despite its simplicity does not ensure providing a correct description of the behavior of magnetorheological materials. The main reason is the occurrence of some interactions inside the material structure. Magnetorheological elastomers combine two main components, the first one is an active ferromagnetic filling in the form of particles, spheres, and flakes of, e.g., iron, and the other one is a matrix, which may be made of, e.g., thermoplastic elastomers, as was the case in this work. The matrix is a binding element, however it simultaneously results in small, local dislocations in the first phase. Together, both elements constitute one uniform structure. The compact filling of space and the adherence of iron particles to each other allows, among others, for some interactions between them. As a result of material deformation on a global scale, friction takes place on the border between the particles of the ferromagnetic filling on a micro-scale. Forces generated as a consequence of these interactions depend on mutual attraction forces, which in turn are induced by an external magnetic field. Based on this type of image, an attempt was made to describe magnetorheological elastomers using dissipative elements that were independent of deformation speed and simultaneously nonlinear. The author previously used the adopted model in works depicting the so-called magnetorheological composites [23,24]. They belong to the same group of Smart Materials activated magnetically. However, they differ in their internal structure and the occurrence of a viscous liquid. The scheme of the adopted model is presented in Figure 5.

It is composed of two parallel connected branches and three elements ($\tau_{o2}$ is a shear yield point, and $G_{1}$ and $G_{2}$ are Kirchhoff’s modulus). The elastic elements cooperate with the ideally plastic element, which takes into account various types of hardening, i.e. kinematic, isotropic and mixed. The constitutive compounds describing the dependence of stress on deformation are presented below:(1)$$\tau =\tau_{1}+\tau_{2}\;\mathrm{and}\;\gamma =\gamma_{1}=\gamma_{2}\;,$$
where:(2)$$\tau_{1}=2G_{1}\gamma \;,$$
(3)$$\tau_{2}=\left\{\begin{array}{ccccc} 2G_{2}\left(\gamma -\gamma_{a}\right)+\tau_{o2} & \mathrm{for} & \dot{\gamma }<0 & \mathrm{and} & \gamma >-\tau_{o2}/G_{2}+\gamma_{a}\\ -\tau_{o2} & \mathrm{for} & \dot{\gamma }<0 & \mathrm{and} & \gamma >-\tau_{o2}/G_{2}+\gamma_{a}\\ 2G_{2}\left(\gamma +\gamma_{a}\right)-\tau_{o2} & \mathrm{for} & \dot{\gamma }>0 & \mathrm{and} & \gamma <-\tau_{o2}/G_{2}-\gamma_{a}\\ \tau_{o2} & \mathrm{for} & \dot{\gamma }>0 & \mathrm{and} & \gamma >-\tau_{o2}/G_{2}-\gamma_{a} \end{array}\right.$$

The adopted model, contrary to the Kelvin–Voigt model, does not consider the dependence on strain rate. It is a rather untypical approach because elastomer models are based mainly on the use of the so-called viscous liquid. In the case of the described magnetorheological elastomers, such dependencies were avoided. The characteristics obtained from the conducted experiments, which allowed observing the interaction of the material at various strain rates, are presented in Figure 6. It was found that, above a certain level of strain rate, stresses in the material state had a constant value. This allows adopting the assumption of an insignificant or non-existent influence of rate or frequency. It is justified when the used magnetorheological elastomer is employed to work with higher frequencies. In the described case, the border point can be assumed to be ${\dot{\gamma }}_{A}=0.025\;\left(1/s\right)$.

## 6. Identification of Model Parameters

The identification of the first two parameters: $G_{1}$ and $\tau_{o2}$ was conducted on the basis of experimental data describing only the material loading process (without unloading). The selected parts of the hysteresis loop were used to find the intersection point with γ-axis and the straight line inclination angle. Such a solution allowed to simultaneously determining these parameters for the model. The linear function approximation was made using the linear regression method based on the least squares method. The consistency of the following function $\tau =\gamma G_{1}+\tau_{o2}$ was analyzed for each loop separately, for both the top and bottom parts of the loop. Calculations were made for all available measurements of all amplitudes and strain rates. Finally, the results were averaged separately for each value of the magnetic field. The consistency criterion being the amount of dissipated energy in one cycle was used to determine parameter $G_{2}$. It was decided to compare the size of loops obtained for the model and the experimental data. The consistency criterion was defined as follows:(4)$$\delta =\Delta W_{E}-\Delta W_{M(G_{2})},$$
where $\Delta W_{E}$ is the area calculated from experimental data, while $\Delta W_{M(G_{2})}$ is the value obtained in the model with set parameter $G_{2}$. The calculations were made in such a way that δ was minimized and then the value of parameter $G_{2}$ was selected. Similar to the first two parameters, the calculations were made separately for each amplitude and deformation speed value and then they were averaged. The procedure was repeated subsequently for the incremental values of a magnetic field. The final identification results are presented in Figure 7. The $G_{1}$ and $G_{2}$ value order was similar. However, the character of their course as a function of a magnetic field was different. The value of parameter $G_{1}$ increased with a magnetic field, while $G_{2}$ decreased. Parameter $\tau_{o2}$ behaved as expected; it grew with the increase in a magnetic field. As explained above, its increase means a rise in interaction forces between ferromagnetic particles inside the material and an increase in friction forces inside the material.

Following the analysis of the correctness of model choice and the conducted identification, a set of hysteresis loops obtained from the model are presented in Figure 8. They were calculated for the same parameters for which the experimental data shown in Figure 4b were recorded. A detailed comparison of individual loops is presented in Figure 9. The differences between the model and experimental loops are not significant and depend on the deformation amplitude.

The set of loops obtained from the model for increasing values of a magnetic field is shown in Figure 10. The model correctly maps changes in loop area and curve inclination during loading, which can be seen by comparing them with the loops presented in Figure 4. The degree to which the model reflects the loops obtained in the experiment was determined by comparing particular waveforms. For this purpose, changes in the areas of hysteresis loops and maximum stress values were analyzed. The differences related to all examined strain amplitudes, magnetic field and frequency did not exceed 10%.

## 7. Summary and Conclusions

The identification process was completed with the determination of the model parameters. Satisfactory results have been obtained for the entire range of the magnetic field used. The course of variation of model parameters has a linear character of dependence on the magnetic field. The correctness of the obtained model was shown in comparison with the results obtained in real studies. Despite the good correlation of experimental data and the model, it should be noted that the adopted equations could not reflect the influence of strain rate changes. The loops obtained for this model are identical, regardless of the frequency of the excitation signals. Hence, the loops presented in Figure 4 are described with only one loop. The limitation of low frequencies did not seem to be a considerable simplification. Regardless of these imperfections, the description of the investigated magnetorheological elastomer obtained using this model can be considered correct. The obtained results are better than in the case of using the viscoelastic Kelvin–Voigt model where the linear viscous function could not correctly reproduce the behavior of the magnetorheological elastomer type for a wide range of strain rate.

## Figures and Tables

**Figure 1 materials-12-00892-f001:**
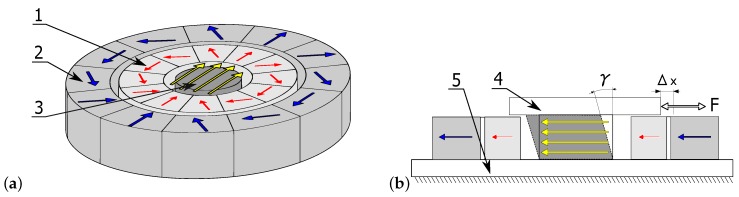
Scheme of magnetorheological elastomer sample placed inside the Halbach matrix: (**a**) view of an array with the sample; and (**b**) cross-section. Notation is presented in the text.

**Figure 2 materials-12-00892-f002:**
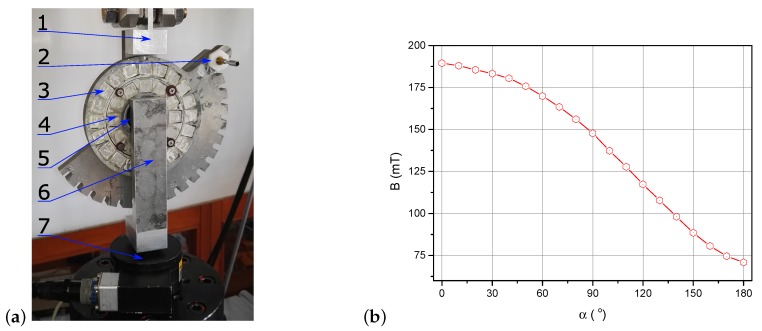
(**a**) Photo of test setup system: 1, upper fix; 2, adjusting lever; 3, external Halbach ring; 4, external Halbach ring; 5, specimen; 6, lower fix; 7, force transducer. (**b**) Characterization of magnetic field dependence on rotation angle α of external ring Halbach matrix.

**Figure 3 materials-12-00892-f003:**
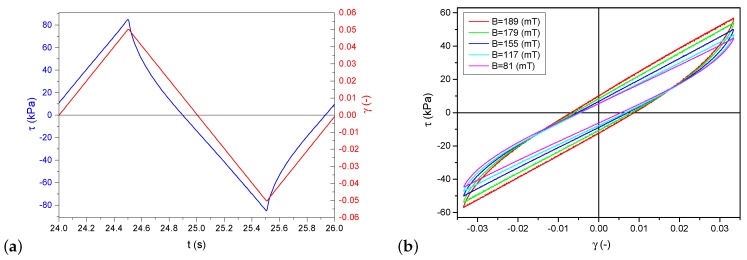
Experimental research results: (**a**) stress and deformation as a function of time for selected value of magnetic field and deformation amplitude; and (**b**) hysteresis loops τ(γ) for a few selected values of magnetic field at constant values of strain amplitude and strain rate.

**Figure 4 materials-12-00892-f004:**
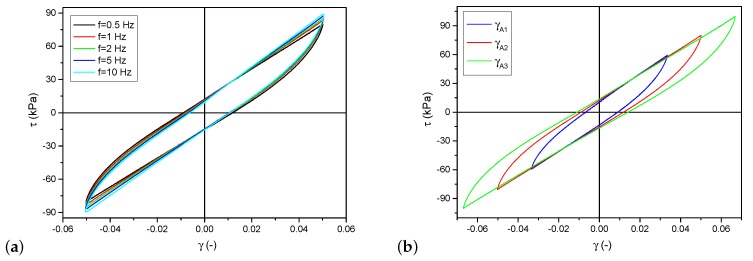
Experimental research results: (**a**) hysteresis loops for various excitation frequencies, at the selected value of magnetic field and deformation amplitude; and (**b**) hysteresis loops for different strain amplitudes, at the chosen value of magnetic field and deformation amplitude.

**Figure 5 materials-12-00892-f005:**
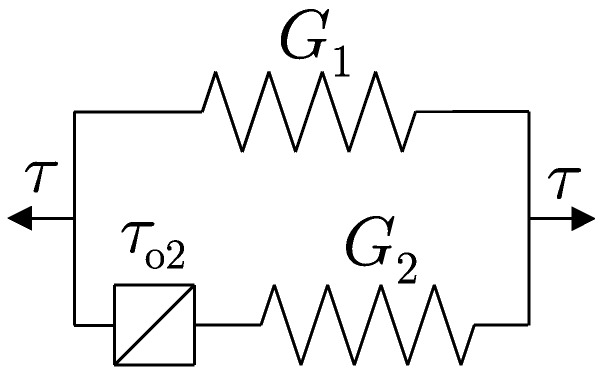
The scheme of elastic–plastic model with kinematic hardening.

**Figure 6 materials-12-00892-f006:**
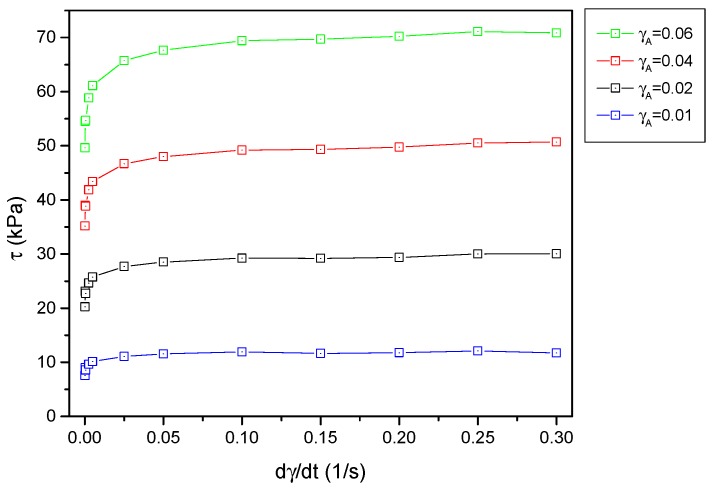
Dependence of stress inside magnetorheological elastomer on strain rate for various strain amplitudes.

**Figure 7 materials-12-00892-f007:**
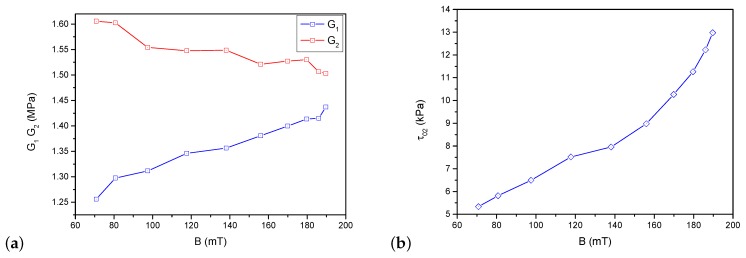
Identification results. Model parameters as a function of magnetic field value: (**a**) stiffness $G_{1}$ and $G_{2}$; and (**b**) yield point $\tau_{o2}$.

**Figure 8 materials-12-00892-f008:**
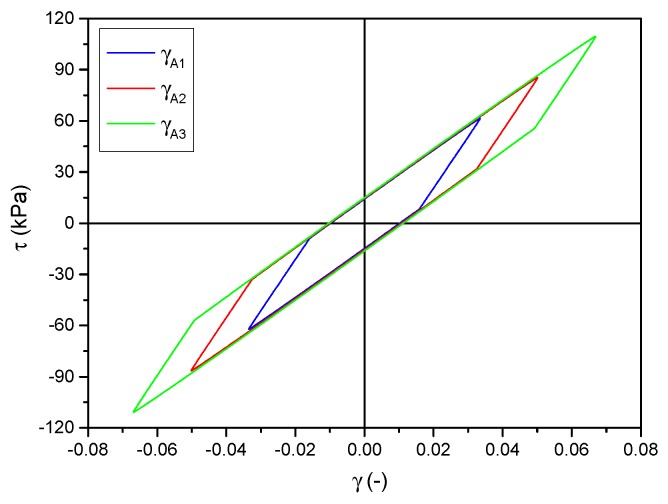
Identification results. Hysteresis loops obtained from the model for three values of deformation amplitude and the set value of magnetic field.

**Figure 9 materials-12-00892-f009:**
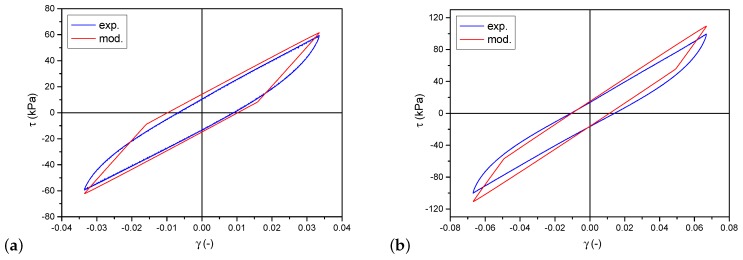
Direct comparison of experimental and model results. Selected hysteresis loops for two different values of strain amplitude: (**a**) $\gamma_{A}=0.035$; (**b**) $\gamma_{A}=0.067$

**Figure 10 materials-12-00892-f010:**
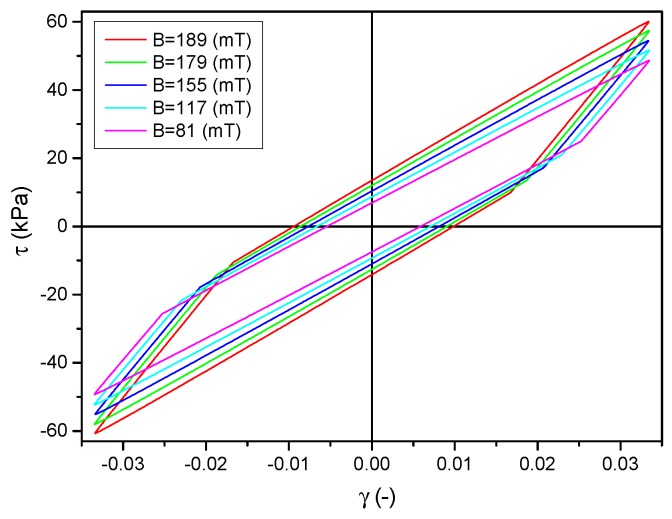
Set of hysteresis loops obtained from the model for subsequently incrementing values of magnetic field.

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
