# Peer review of "Modeling of Magnetorheological Elastomers Using the Elastic–Plastic Model with Kinematic Hardening"

_materials, 2019, doi:10.3390/ma12060892_

Reviewer 1 Report

1. In line 68, the author should explain the physical meaning of Fig.2. Why rotation angles change magnetic fields?

2. In line 70, "moda" is a wrong word.

3. In line 78, strain rate is defined as "gamma dot". However, in Figs. 3, 4, 8, etc., strain rates seem to be expressed by the author as "gamma star", which is confusing.

4. What does "i" represent in Equation (1)?

5. The unit of deformation amplitude is mm, etc. The unit of strain rate is 1/s. In line 79 and Fig.3(a) caption, "deformation amplitues" are probably wrong phrases, and maybe should be written as "strain amplitudes".

6. Again, in Fig.6 caption, "deformation amplitudes" probably should be written as "strain amplitudes".

7. Again, in Fig.9, the use "gamma star" symbol is confusing. The use of "deformation amplitudes" probably should be written as "stain amplitudes", now that deformation units are mm, for example.

Author Response

Thank you very much for a thorough review and comments to my article. They gave me an additional look at the work and its meaning.

1. In line 68, the author should explain the physical meaning of Fig.2. Why rotation angles change magnetic fields?

Added explanation:

Changing the rotation angle between the rings causes the strengthening or weakening of the magnetic field inside it. This is due to the total sum of the magnetic field of all individual magnets that make up the matrix.

2. In line 70, "moda" is a wrong word.

Corrected

3. In line 78, strain rate is defined as "gamma dot". However, in Figs. 3, 4, 8, etc., strain rates seem to be expressed by the author as "gamma star", which is confusing.

I corrected notation. In Fig 3,4,8,9,10 there is only strain on x axis.

4. What does "i" represent in Equation (1)?

Corrected. There have to be “and”

5. The unit of deformation amplitude is mm, etc. The unit of strain rate is 1/s. In line 79 and Fig.3(a) caption, "deformation amplitues" are probably wrong phrases, and maybe should be written as "strain amplitudes".

6. Again, in Fig.6 caption, "deformation amplitudes" probably should be written as "strain amplitudes".

7. Again, in Fig.9, the use "gamma star" symbol is confusing. The use of "deformation amplitudes" probably should be written as "stain amplitudes", now that deformation units are mm, for example.

I corrected all “deformation amplitudes” to “strain amplitudes”

Reviewer 2 Report

The article describes a new way of modelling the properties of magneto mechanical properties of magnetorheological elastomers using parallel elastic elements. The identification of the parameters is done on the basis of experimental data. This is a real added value of the paper even if the experimental/theoretical comparison should be improved.

Despite many corrections which must be done, the paper presents an original work which can be published in Material Journal after minor (but numerous) revisions.

The corrections concern the presentation of the paper together with the description of the experimental and theoretical results. The list of improvements is as follow:

Introduction:

-        The introduction should be developed, line 18: this sentence should be extended, why this kind of material should be used in energy dissipation and vibration damping? Could this material be used as an actuator?

-        Line 13 “analysis”

-         line 15 “characteristics”

-        Line 26: the example should be detailed

-        Line 45: Explain the structure of the paper.

Paragraph 3:

-        How the moving top plate is attached to the sample?

-        Alpha angle should appear on figure 1,

Paragraph 4:

-        Line 74: “characteristic"

-        Figure 3-a, what is the gamma angle unit?

-        Figure3-b, the magnetic field is called B and not H,

Paragraph 5:

-        Line 87: “at it is the case”

-        Figure 5: is tau a symmetrical variable

-        G1 and G2 should be clearly defined,

Paragraph 6:

-        “Without unloading” means “with loading”?

-        The consistency criterion is defined but not used in the paper this must be corrected.

-        Figure 10: Magnetic field should be B

Conclusion:

The conclusion must be rewritten using the analysis of the consistency criterion.

Moderate English change are required.

Author Response

Thank you very much for a thorough review and comments to my article. They gave me an additional look at the work and its meaning.

The article describes a new way of modelling the properties of magneto mechanical properties of magnetorheological elastomers using parallel elastic elements. The identification of the parameters is done on the basis of experimental data. This is a real added value of the paper even if the experimental/theoretical comparison should be improved.

Despite many corrections which must be done, the paper presents an original work which can be published in Material Journal after minor (but numerous) revisions.

The corrections concern the presentation of the paper together with the description of the experimental and theoretical results. The list of improvements is as follow:

Introduction:

-        The introduction should be developed, line 18: this sentence should be extended, why this kind of material should be used in energy dissipation and vibration damping? Could this material be used as an actuator?

I added an explanation to the introduction. “Such applications are possible because they contain dissipation energy mechanisms that can be controlled and changed by a magnetic field. These materials can also be used as actuators, which requires taking into account their magnetostrictive properties”.

-        Line 13 “analysis”

Corrected

-         line 15 “characteristics”

Corrected

-        Line 26: the example should be detailed

The examples have been detailed.

-        Line 45: Explain the structure of the paper.

This paper has been divided into sections. In the initial part, the MRE material and samples for testing were described. Next, the description of the measurement stand and the test procedure was presented. The most important part of the article is the model and procedure for the identification of its parameters, taking into account the influence of the magnetic field. The work was completed by comparing the hysteresis loops obtained from the model and experimental data.

Paragraph 3:

-        How the moving top plate is attached to the sample?

Specimens were glued to the plates. I used Loctite 480, a cyanoacrylate glue dedicated to rubber bonding.

-        Alpha angle should appear on figure 1,

Alpha angle is shown on Fig 1b.

Paragraph 4:

-        Line 74: “characteristic"

-        Figure 3-a, what is the gamma angle unit?

Gamma angle is dimensionless, denoted as (-). I calculate gamma as the ratio of displacement upper plate to distance between plates.

-        Figure3-b, the magnetic field is called B and not H,

Corrected.

Paragraph 5:

-        Line 87: “at it is the case”

Corrected.

-        Figure 5: is tau a symmetrical variable

Yes it is.

-        G1 and G2 should be clearly defined,

tau_o2 is a shear yield point, G_1 and G_2 are Kirchhoff’s modulus.

This information is also added to the text.

Paragraph 6:

-        “Without unloading” means “with loading”?

This statement means that for the analysis of the first two parameters (G1, tauo2), only the section obtained during loading the sample was taken into account.

-      The consistency criterion is defined but not used in the paper this must be corrected.

This criterion has been used! I used in this paper to calculate the G2 parameter. The calculations of delta have not been shown because they do not bring anything relevant to work.

-        Figure 10: Magnetic field should be B

Corrected

Conclusion:

The conclusion must be rewritten using the analysis of the consistency criterion.

As mentioned above, the delta criterion was only used to determine the G2 parameter. The overall assessment of the model loop and experimental data was only made by showing a comparison of the hysteresis loop as shown in Figures 9a and 9b. A more accurate analysis of the correctness of the model requires a comparison of results in a wide range of parameter variability, which may be a contribution to the next article.

Moderate English change are required.

The article has been read and corrected additionally by a native speaker.

Reviewer 3 Report

Thanks for inviting me to review this manuscript. This research investigates the parameters of elastic-plastic model for magnetorheological elastomers through experimental testing. The determined parameters are then used to model the hysteresis loops of the elastomers. The organization of the manuscript is clear. However, the significance of the research is limited. I did not see enough technical contribution. Limited experimental testing was conducted using well established methods, and the analysis of the testing data is straightforward. Also, the language is not easy to follow. So, I cannot recommend it for publication in the current form. 

Here are some comments and questions that might be useful for further improvement:

(1) In the Introduction, highlight the technical contributions of the research, and point out the significance and potential impacts of the research. This is my major concern. 

(2) A lot of details of the experimental testing is missing. Please clarify the details of the specimens, test set-up, instrumentation, test protocol, etc. Please add photos of the specimens and test set-up. 

(3) The data analysis and discussions are too straightforward and do not convey enough new knowledge. Please consider to add more analyses to enhance the significance of the study. 

(4) Language is readable but contains a lot of typos and ambiguous sentences. Please polish the language. 

Author Response

Thank you very much for a thorough review and comments to my article. They gave me an additional look at the work and its meaning.

Thanks for inviting me to review this manuscript. This research investigates the parameters of elastic-plastic model for magnetorheological elastomers through experimental testing. The determined parameters are then used to model the hysteresis loops of the elastomers. The organization of the manuscript is clear. However, the significance of the research is limited. I did not see enough technical contribution. Limited experimental testing was conducted using well established methods, and the analysis of the testing data is straightforward. Also, the language is not easy to follow. So, I cannot recommend it for publication in the current form. 

Here are some comments and questions that might be useful for further improvement:

(1) In the Introduction, highlight the technical contributions of the research, and point out the significance and potential impacts of the research. This is my major concern. 

The main purpose of this work was to show the possibilities of modeling magnetorheological elastomers with the use of a new model. The focus was not on describing the technical aspects. Only the most important parameters and details of construction and testing were included. The author's previous works contain much more information about the method of testing and technical detail. This information’s has been added to work.

(2) A lot of details of the experimental testing is missing. Please clarify the details of the specimens, test set-up, instrumentation, test protocol, etc. Please add photos of the specimens and test set-up. 

Information about the specimens has been included in Paragraph 2. A photo of the research system has been added to the work (Fig 2a).

(3) The data analysis and discussions are too straightforward and do not convey enough new knowledge. Please consider to add more analyses to enhance the significance of the study. 

The conclusion paragraph has been changed and expanded.

(4) Language is readable but contains a lot of typos and ambiguous sentences. Please polish the language. 

The article has been read and corrected additionally by a native speaker.

Reviewer 4 Report

The paper presents a modeling of magnetorheological elastomers using elastic-plastic model with kinematic hardening.

The manuscript can be considered for publication after accounting for comments proposed below.

major comments: 

The author shows that the model well describes experimental results for strain rates above 0.025s-1. However, this model is only applied to a strain rate range where stress and strain rate are independent for a given amplitude. Could you provide more evidence on the interest of such model as compared to those present in the literature? More specifically, could you show the interest of this model by comparing it not only with the experiments but also if possible with the commonly used Kelvin-Voigt model. 

Minor changes:

please correct the writing mistakes: line 13: replace "analysis" by "analysis"; line 14 replace "choosed" by chosen", line 34: replace "streass" by "stress", line 59: replace "set constant" by "set of constant", line 137: replace "with regardless" by "regardless", line 81: replace "experimental experience" by "experiments".

Author Response

Thank you very much for a thorough review and comments to my article. They gave me an additional look at the work and its meaning.

The paper presents a modeling of magnetorheological elastomers using elastic-plastic model with kinematic hardening.

The manuscript can be considered for publication after accounting for comments proposed below.

major comments: 

The author shows that the model well describes experimental results for strain rates above 0.025s-1. However, this model is only applied to a strain rate range where stress and strain rate are independent for a given amplitude. Could you provide more evidence on the interest of such model as compared to those present in the literature? More specifically, could you show the interest of this model by comparing it not only with the experiments but also if possible with the commonly used Kelvin-Voigt model. 

Due to the limitations in the size of the work, there is no comparison of the selected model with the K-V model. However, such analyses were carried out at the preliminary research stage. Parameters for the K-V model were determined for a whole series of amplitudes and strain rates. The values of elasticity and viscosity changed even by several dozen percents for individual measurements. The values of these parameters were then averaged. For such determined elasticity and viscosity differences in the size of experimental loops and from the model reached over several hundred percents. On this basis, the K-V model was considered unsuitable. In the literature, many people identify the parameters of the K-V model and then depend them on the speed of deformation (the frequency of excitation). However, this is a wrong approach because it means that the model is only valid for one speed.

Minor changes:

please correct the writing mistakes: line 13: replace "analysis" by "analysis"; line 14 replace "choosed" by chosen", line 34: replace "streass" by "stress", line 59: replace "set constant" by "set of constant", line 137: replace "with regardless" by "regardless", line 81: replace "experimental experience" by "experiments".

Corrected.

Round  2

Reviewer 3 Report

The author has addressed my major concerns. Here are some minor suggestions:

(1) Please spell out "exp" and "mod".

(2) What is the unit of gamma in Figs. 8-10?

(3) Please list key findings as bullets, rather than a summary. Some conclusions are not supported by the data, so please remove. For instance, "it can be used, e.g. in control systems and vibration control". Only give the conclusions that are supported. 

Author Response

(1) Please spell out "exp" and "mod".

I changed the signs from exp to E and mod to M

(2) What is the unit of gamma in Figs. 8-10?

Gamma angle is dimensionless, denoted as (-). I calculate gamma as the ratio of displacement upper plate to distance between plates.

(3) Please list key findings as bullets, rather than a summary. Some conclusions are not supported by the data, so please remove. For instance, "it can be used, e.g. in control systems and vibration control". Only give the conclusions that are supported. 

I removed not supported conclusion.

Reviewer 4 Report

The author of the manuscript accounted for all reviewers comments, which is appreciable. The scientific quality ad clarity of the manuscript has been improved significantly.

Overall, I noted that all sections were well improved, especially the introduction, experimental section. More specifically, the advantage of the developed model as compared to the K-V model is clear now. Model frame and assumptions are also clearly presented.

The manuscript is suitable for publication in Materials.

One minor correction was detected: line 21: please remove ")"

Author Response

One minor correction was detected: line 21: please remove ")"

Corrected.